# The Impacts of Urban Configurations on Outdoor Thermal Perceptions: Case Studies of Flat Bandar Tasik Selatan and Surya Magna in Kuala Lumpur

Lin Yola [1,*] , Timothy O. Adekunle [2] and Olutobi G. Ayegbusi [3]

1 Urban Studies Department, School of Strategic and Global Studies, Universitas Indonesia, Jakarta 10430, Indonesia
2 School of Architecture, College of Architecture & Planning, University of Utah, Salt Lake City, UT 84112, USA
3 Faculty of Built Environment, Linton University College, Mantin 71700, Malaysia
* Correspondence: lin.yola@ui.ac.id; Tel.: +62-21-392-9717

**Abstract:** This study assesses the impacts of urban configurations on thermal perceptions in Flat Bandar Tasik Selatan (FBTS) and Surya Magna (SM) in Kuala Lumpur, Malaysia. It aims to understand the impacts of urban configurations on thermal perceptions in outdoor spaces. The study addresses the following research questions: (1) Do urban configurations influence outdoor comfort? (2) Do urban configurations also have significant impacts on thermal perceptions? We mapped out the sites to understand their configurations. The research considered on-site measurements of the environmental conditions and carried out modelling and simulations (ENVI-met V3.1) of the sites. Mathematical models (Wet-Bulb Globe Temperature (WBGT), Universal Thermal Climate Index (UTCI), and Standard Effective Temperature (SET)) were used to determine the thermal indices and the impact of the urban configurations on outdoor comfort. The thermal indices varied from 25.44 to 34.75 °C. In terms of the main contribution of this work, the results show that in hot and humid climate regions, urban configurations plus other design variables (e.g., orientation towards the Sun's path) and environmental parameters influence occupants' comfort and perceptions. Our findings show that high solar radiation and the need for a better thermal environment in hot and humid climates are the contributing factors for developing alternative urban configurations.

**Keywords:** urban configurations; thermal perceptions; canyon; outdoor thermal comfort; microclimate; site models

## 1. Introduction

Numerous studies have assessed the indoor [1,2], and outdoor [3–7] thermal comfort of occupants in different climates. Some studies have examined the outdoor comfort of city dwellers [8–10] and the impact of urban configurations on the performance of street canyons [6]. Previous research [7,11] has also examined the impact of Urban Heat Island (UHI) in metropolises. In previous investigations [6,12,13], UHI has been described as a critical phenomenon impacting urban configurations in many locations. Additionally, UHI has led to increasing temperatures in urban areas compared to the surrounding rural environment.

Prygou et al. [4] explained that elevated temperatures in urban areas can lead to additional energy costs because of increasing energy demand for cooling, etc., thereby causing additional demand on the energy grid during extreme conditions. Ambient temperatures can affect people's health and their overall wellbeing [5]. Comfort and health are also critical indicators to describe people's wellbeing within the thermal environment [14,15]. Therefore, more studies have been assessing outdoor comfort.

In many regions, climatic data often show that cities are characterised by elevated temperatures, humidity, and decreasing wind speeds [6]. Moreover, in hot and humid

regions, the impact of UHI is more significant than in other regions across the globe. In Kuala Lumpur, Malaysia, this statement is also true—especially during the warm season. A recent report revealed that approximately 33% of the world's population resides in cities [6]. Kuala Lumpur has an estimated population of 8.4 million as of the first quarter of 2022 [16]. According to the Köppen Climate Classification, Kuala Lumpur is classified as Af (i.e., tropical, rainforest climate, fully humid). In addition to being hot and humid, Kuala Lumpur is also characterised as a tropical city with clear skies and high solar radiation.

As reported in previous investigations, UHI is also influenced by urban configurations, buildings, and materials [11,17]. Well-designed urban spaces can enhance outdoor occupants' comfort and, thus, promote health and overall wellbeing [7,18]. Well-designed outdoor spaces can also lead to frequent use of such spaces [19] and improve the levels of interaction among the users [20]. Additionally, well-designed and thermally comfortable spaces can reduce people's vulnerability to elevated temperatures and thermal stress [21–24].

As a result, the present study assesses and discusses the impact of urban configurations on thermal perceptions in two locations in Kuala Lumpur, Malaysia. The novelty of this research is to evaluate the impact of urban configurations on thermal perceptions and heat indices in the context of tropical hot and humid urban spaces. The research questions are as follows: (1) Do urban configurations influence occupants' comfort in outdoor spaces? (2) Do urban configurations have significant impacts on thermal perceptions in urban spaces? The research objectives are as follows:

(a)   To assess the impact of urban configurations on outdoor comfort.
(b)   To examine thermal perceptions and discuss the impact in the study location
(c)   To discuss the applications and recommendations based on the outcomes of the research.

This study explores different research techniques to assess outdoor comfort in the study location.

## 2. Literature Review

Previous research has discussed different issues that can impact occupants' comfort in urban areas [25–29]. These issues include urban microclimate [28], urban thermal comfort [29], anthropogenic heat, and energy consumption [28,30]. According to Yola [18], urban areas that are classified as uncontrolled, climatically unresponsive environments are associated with such issues and characterised by blocked daylight and urban wind due to high-rise buildings.

While different elements in the built environment have been found to impact microclimates [28], the adoption of appropriate passive design concepts is necessary to achieving a better microclimate [31]. Researchers have examined the impacts of urban greenery [32], cool urban construction materials [33], the application of water bodies, and the combination of these strategies on the overall urban microclimate [34–38]. From existing research, the present study notes that climatically responsive urban environments should consider microclimate modifications. Moreover, a climatically responsive urban environment should be sustainable. Yola et al. [18] discussed the concept of climatically responsive urban configuration in a tropical context. Table 1 summarises existing research on the impact of urban configurations on microclimate and thermal comfort in the climatic context of different study areas, including the four-season, tropical, and hot–arid regions. The detail matrix of review—including the study area, types of building use and configurations, physical features, climatic features, and findings—was presented in a previous study [39], and is compared with the present climatic responsive urban configuration study.

**Table 1.** Studies on the impact of urban configurations on microclimate and thermal comfort.

| Research Areas | Researchers, Year |
|---|---|
| Impact of urban configurations on microclimate | Oke, 1979 [40]; Gupta, 1984 [41]; Arnfield, 1990 [42]; Givoni, 1998 [43]; Elhanas, 2003 [44]; Ng, 2010 [45]; Erell et al., 2011 [46]; Priyadarsini and Wong, 2011 [47]; Yuan and Ng, 2012 [48]; Allegrini et al., 2015 [49]; Kariminia et al., 2015 [50]. |
| Impact of urban configurations on thermal comfort | Algeciras et al., 2016 [51]; Taleghani et al., 2015 [52]; Ghaffarianhoseini et al., 2015 [53]; Abdallah, 2015 [54]; Ndetto and Matzarakis, 2013 [55]; Martins et al., 2012 [56]; Krüger et al., 2011 [57]; Herrmann and Matzarakis, 2010 [58]; Johansson, 2006 [59]. Muhaisen and Abed, 2014 [60]; Alznafer, 2014 [61]; LSE Cities, 2014 [62]; Futcher and Mills, 2013 [63]; Dorer et al., 2013 [64]; Abed, 2012 [65]; Creswell-Wells, 2012 [66]; Hachim et al., 2011 [67]. |

Pioneering research relating to urban energy balance and Urban Heat Island (UHI) was published by Oke [39,68]. The investigations explained that maximum Urban Heat Island (UHI) in urban spaces is generated by a high Height-to-Width (*H/W*) aspect ratio and small Sky View Factor (*SVF*). The parameters can be computed using Equations (1) and (2). In this study, these equations are referred to as Oke's model [68]. Moreover, in this context, the height (*H*) refers to the vertical obstruction, and the width (*W*) refers to the distance between the vertical obstructions. The Sky View Factor (*WGBT*) is defined as the ratio of ground surface exposure to the unobstructed sky hemisphere. Therefore, a high UHI was mostly recorded in high-density cities with high-rise buildings. In this context, the intensity of UHI was determined by configurations of urban canyon space [69]. In the present study, urban canyon is examined as a repetitive configuration in the urban area. Though Oke's model on in tensity of $dT_{max}$ [68] discussed UHI within urban canyon spaces, this investigation assesses the model and thermal indices in alternative settings of urban configuration.

$$dT_{max} = 7.45 + 3.97 * ln\left(\frac{H}{W}\right) \tag{1}$$

$$dT_{max} = 15.27 + 13.88 * SVF \tag{2}$$

Existing research has evaluated occupants' comfort in outdoor spaces using different thermal indices, such as the Wet-Bulb Globe Temperature (WBGT) [7], Universal Thermal Climate Index (UTCI) [7,70,71], Standard Effective Temperature (SET) [71,72], Predicted Mean Vote (PMV) [71], Physiological Equivalent Temperature (PET) [71,73], and others [74–81]. Generally, in outdoor thermal spaces, thermal comfort is influenced by the dominant microclimate variables, such as wind velocity, etc. [69,75]. These parameters are the criteria to evaluate the individual comfort state in any given climate. In this study, WBGT, SET, and UTCI are explored to understand the thermal indices in the study locations. Wet-Bulb Globe Temperature (*WGBT*) was first used to assess cases of thermal stress in military facilities [82]. This index takes into consideration parameters such as ambient conditions, clothing insulation, and human activities [83]. Equations (3) and (4) can be applied to calculate *WBGT*:

$$WBGT = 0.7T_W + 0.2T_G + 0.1T_W \tag{3}$$

$$WBGT = 0.7T_W + 0.3T_{db} \tag{4}$$

In the above equations, WBGT is the Wet-Bulb Globe Temperature (°C), *Tw* is the wet-bulb temperature (°C), $T_G$ is the black globe temperature (°C), and $T_{db}$ is the dry-bulb temperature (°C). According to Lemke and Kjellstrom [84], $T_G$ is also noted as the dry-bulb temperature when assessing thermal indices in shaded areas. Standard Effective Temperature (*SET*) is defined as the combined effect of the standard heat transfer and the coefficients of evaporative heat transfer, the portion of the wetted skin surface, the water vapour pressure on the skin, and the saturated water vapour pressure [72]. This thermal index is computed by estimating the thermo-physiological behaviour of human beings (Equation (5)).

$$H_{sk} = h_s(t_{sk} - SET) + wh_{s,e}(p_{s,sk} - 0.5p_{SET}) \tag{5}$$

where $H_{sk}$ is the heat loss from skin (W/m$^2$), *hs* is the coefficient of standard heat transfer (W/m$^2$·°C), $t_{sk}$ is the skin temperature (°C), w is the portion of the wetted skin surface, *hs,e* is the coefficient of standard evaporative heat transfer (W/m$^2$ kPa), *ps,sk* is the water vapour pressure on the skin—usually estimated to be that of saturated water vapour at tsk (kPa)—and pSET is the saturated water vapour pressure at *SET* (kPa). When operative temperatures are required to compute *SET* values, Equation (6) can be applied to calculate the parameter within the thermal environment for metabolic rates that may exceed 1.3 met. The occupants may be in areas that have direct sunlight and are exposed to air velocities higher than 0.10 m/s [85]. In the equation, to is the operative temperature, tmr is the mean radiant temperature, ta is the air temperature, and v is the air velocity, which also explains the relationships among the variables investigated in this study. In this study, the mean radiant temperature calculation was embedded in the ENVI-met model.

$$t_o = \frac{\left(t_{mr} + \left(t_a \ x \ \sqrt{10v}\right)\right)}{1 + \sqrt{10v}} \tag{6}$$

The Universal Thermal Climate Index (UTCI) is described as a human biometeorology variable that is used to evaluate the association between the outdoor environment and human wellbeing [86]. It can be computed by applying a sixth-order polynomial calculation, as discussed in previous research [87]. The detailed information on background, literature review, and how to compute the index (UTCI) has been discussed in prior investigations [7,86–89].

The present study considers WBGT and UTCI concurrently to understand the range of thermal indices to which people may be vulnerable within the thermal environment. Moreover, considering different thermal indices will provide better understanding and opportunities to compare those indices and connect them with comfort indices within the study location. Past investigations have also considered WBGT and UTCI concurrently to evaluate thermal indices in different environments [2,7,22,88,89]. As previously outlined in existing research [88,89], Table 2 summarises the similarities and differences between the indices.

**Table 2.** Temperature bands and classes of thermal indices for WBGT and UTCI.

| WBGT Indices | Period | UTCI Indices | * Classes of the Indices |
|---|---|---|---|
| Temperature less than 28.6 °C | Less than 60 min per hour | Temperature less than −40 °C | Extreme cold stress |
| | | | Very strong cold stress |
| Temperature equals 29.3 °C | Less than 45 min per hour | Temperature equals −27 °C | |
| | | | Strong cold stress |
| Temperature equals 30.6 °C | Less than 30 min per hour | Temperature equals −13 °C | |
| | | | Moderate cold stress |
| Temperature equals 31.8 °C | Less than 15 min per hour | Temperature equals 0 °C | |
| | | | Slight cold stress |
| | | Temperature equals 9 °C | |
| | | | No heat stress |
| | | Temperature equals 26 °C | |
| Temperature that exceeds 38 °C | Less than 0 min per hour | | Moderate heat stress |
| | | Temperature equals 32 °C | |
| | | | Strong heat stress |
| | | Temperature equals 38 °C | |
| | | | Very strong heat stress |
| | | Temperature that exceeds 46 °C | Extreme heat stress |

* Bands of temperatures at which people can experience heat stress.

## 3. Materials and Methods

### 3.1. Description of Case Studies

In this study, two empirical sites were evaluated for analysis. Firstly, the research considered the scenario of empirical urban configurations with the canyon direction of east–west as well as in the north–south direction. Both sites are in high-rise residential building zones in Kuala Lumpur, Malaysia. These sites were selected because they have the same type of empirical urban configuration—in this case, a courtyard canyon. Additionally, the sites are within the same geographical region with similar climatic conditions. Because this study investigated sites in a tropical region, the empirical sites were considered based on their canyon direction being towards the path of the Sun. The first site is Flat Bandar Tasik Selatan (FTBS), which is situated with the canyon direction parallel to the path of the Sun (east–west). The second site is Surya Magna (SM), with the canyon direction perpendicular to the path of the Sun (north–south).

The simulated areas considered in this study referred to the two empirical sites and their immediate surroundings. However, the simulated areas covered the investigated urban configuration with their outdoor microscale environment. The site scenarios (FBTS parallel with the Sun's path and SM perpendicular to the Sun's path) are detailed in Figure 1. The ENVI-met model domain of this study includes the setting of the grid and the geographical properties. The grid cell in the main domain area includes $\Delta$x, $\Delta$y, and $\Delta$z, in metres. This study applied grid cells of x = 210, y = 210, and z = 30, with a nesting grid of 6. The grid cells were set to a small size to maximise the investigation resolution of the microscale thermal indices in the target area The target area of the model domain presents the significant features of the microscale sites—the building blocks, vegetation (i.e., trees and grasses), and surface properties (x and y horizontal dimensions) such as concrete, asphalt, and loamy soil.

This study used Oke's model (Equations (1) and (2)) to examine the role of the Height-to-Width (H/W) aspect ratio and Sky View Factor (SVF). We sought to investigate and validate Oke's model in the context of a tropical, hot and humid region (as it was empirically formulated from a study in a four-season climatic region). To this end, we investigated the building blocks in the existing scenario (i.e., courtyard canyon) and examined the hypothetical scenario (i.e., courtyard, U, and canyon), where all of them complied with Oke's model physical canyon geometry. The Height-to-Width (H/W) aspect ratio remained constant in this study, while the Sky View Factor (SVF) value varied from one urban configuration to another (see Figure 1 and Table 2). However, based on field observation, the pattern of urban microclimate data (i.e., air temperature, relative humidity, and air velocity) in the outdoor courtyards of the building blocks appeared to be impacted by the shadow effects of the vertical obstruction surrounding the existing site. Therefore, to investigate the impact of changes in the Sky View Factor (SVF) value (see Oke's model) on the thermal indices, a change in the urban configuration (i.e., the hypothetical scenario) was examined by ENVI-met simulation in this study.

The features of the two sites and their surrounding environment (with the target urban configuration simulation area marked in red) are presented in Table 3. Similarly, four urban configurations in RayMan models with varying Sky View Factor (SVF) values are also presented as a subfigure in Table 3.

Urban microclimate and thermal comfort are two important measures in assessing urban energy budgets. However, studies frequently highlight urban thermal comfort as the consequence of modification of the urban microclimate [90–93]. In addition to the causal relationship between the two measures, a significant gap is mainly highlighted in the parameters and methods of assessment. Urban microclimate is measured using empirical climatic variables, while urban thermal comfort is assessed according to climatic variables and dominant human effects on the thermal environment. Therefore, this study discusses both microclimate and thermal comfort indices. Air temperature as the main variable of microclimate and mean radiant temperature as the main thermal comfort index are compared in this study to pinpoint the gap. Air temperature is the most important climatic

feature and the most commonly used parameter in thermal environment studies [94,95], as it is commonly used to measure the air in outdoor and indoor environments. Meanwhile, mean radiant temperature (Tmrt) is the most important variable in thermal comfort studies or human energy balance [96–100]. In addition to the complexity of investigating outdoor thermal comfort indices, studies show a significant relationship between Tmrt and outdoor thermal comfort indices such as PMV, PET, and SET [99].

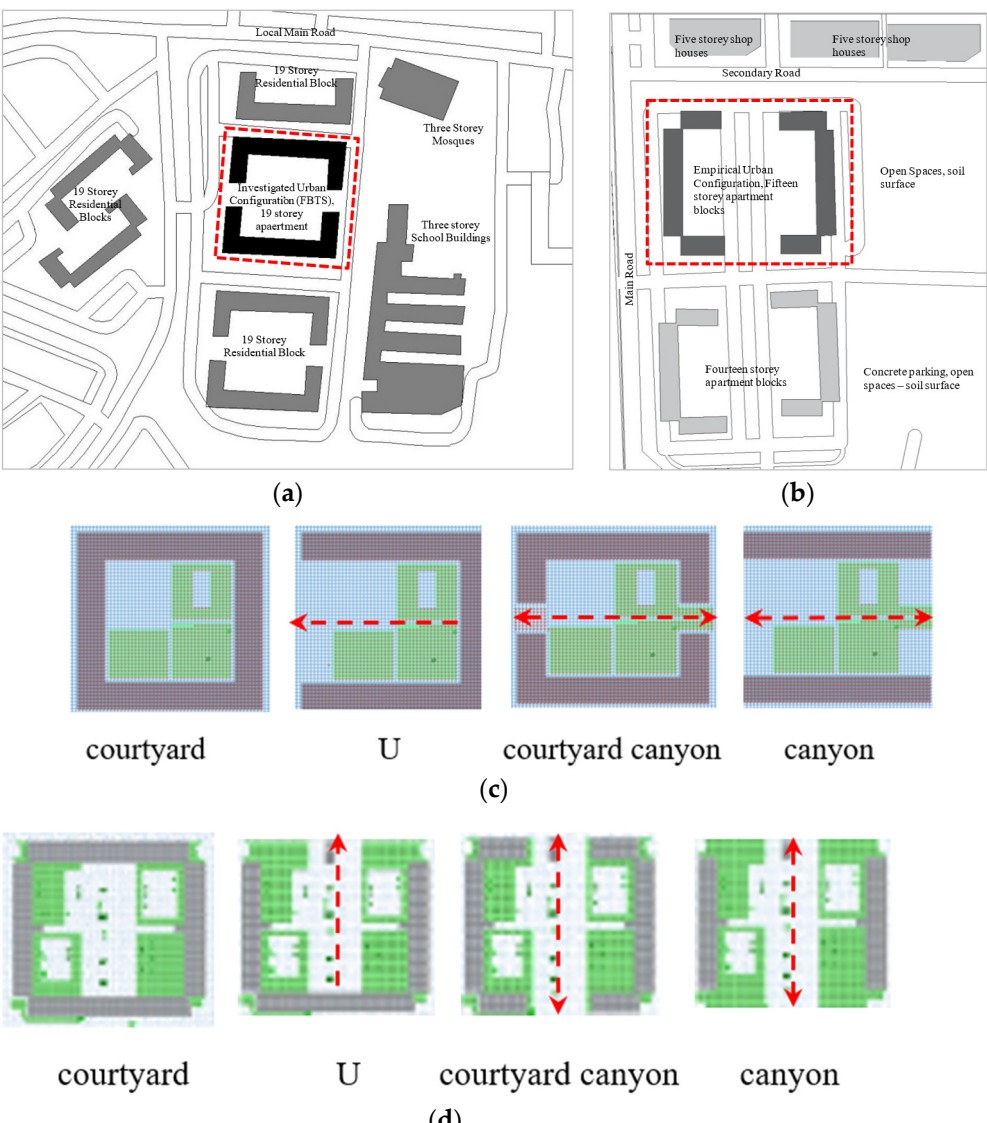

**Figure 1.** The site map with the existing configuration and surrounding situation of FBTS (**a**) and SM (**b**), and the target area of ENVI-met urban configuration simulation at both sites—FBTS (**c**) and SM (**d**). The red arrows show the canyon direction parallel and perpendicular to the path of the Sun.

**Table 3.** Summary of the main features of the case studies.

| Case Study/Description | Configurations without Canyon Feature | Configurations with Canyon Feature |
|---|---|---|
| | ENVI-met model of four urban configurations situated at the Flat Bandar Tasik Selatan site (highlighted in red) | |
| | 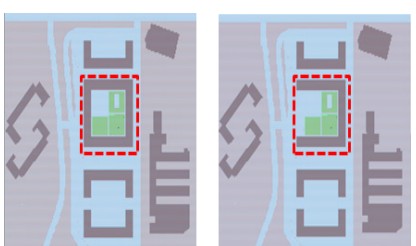 | 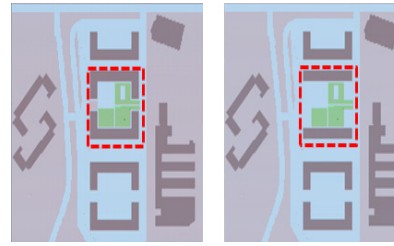 |
| | (a) Courtyard (hypothetical)—left <br> (b) U (hypothetical)—right | (a) Courtyard canyon (existing)—left <br> (b) Canyon (hypothetical)—right |
| Case 1—Flat Bandar Tasik Selatan (FTBS): The site is located in the Cheras residential area, Kuala Lumpur. It consists of 19-storey apartments (60 m height) with the surrounding residential blocks of different heights. The outdoor space is usually shaded by a change in the Sun's altitude, and it mainly functions as a hub for the residents' social outdoor activities. The outdoor space surface consists of grass, trees, concrete pavement, and asphalt. | RayMan-generated fisheye SVF hemispheres of four urban configurations in the east–west canyon direction (Flat Bandar Tasik Selatan) | |
| | 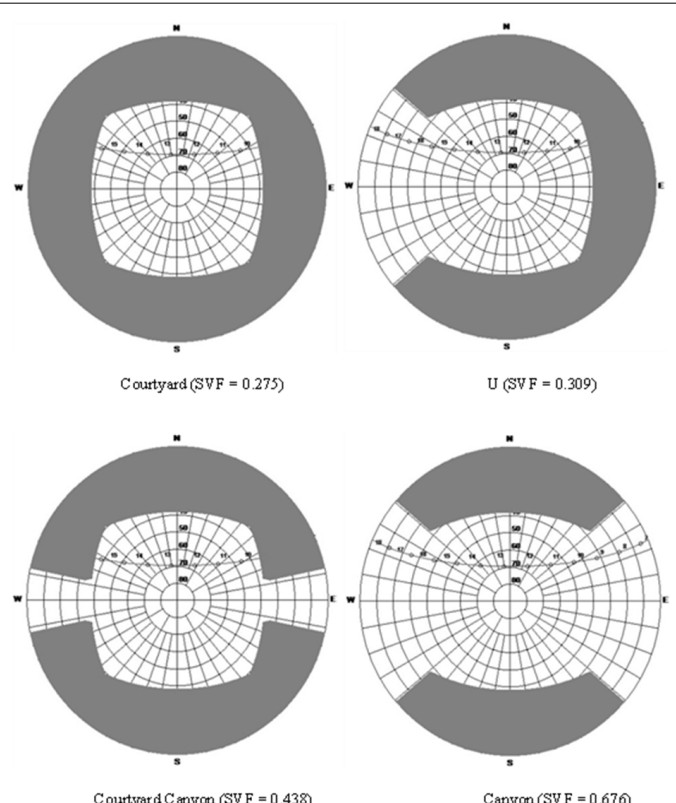 | |

**Table 3.** *Cont.*

| Case Study/Description | Configurations without Canyon Feature | Configurations with Canyon Feature |
|---|---|---|
| | ENVI-met model of four urban configurations situated at the Surya Magna site (highlighted in red) | |

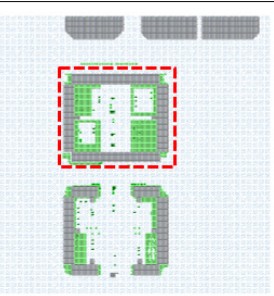

(a) Courtyard (hypothetical)—top

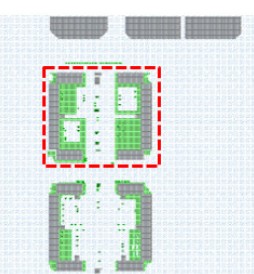

(a) Courtyard canyon (existing)—top

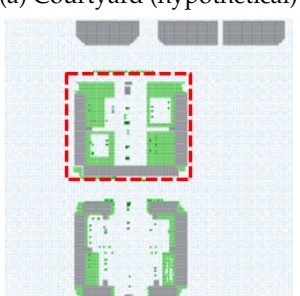

(b) U (hypothetical)—below

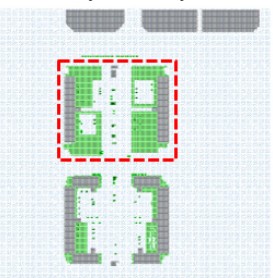

(b) Canyon (hypothetical)—below

Case 2—Surya Magna (SM): Surya Magna (SM) is located in Kepong, Kuala Lumpur. It is a 15-storey apartment (45 m) surrounded by dense residential and commercial areas. The outdoor open space is mainly used for sitting and playground areas. The ground surface includes grass, dense trees, and concrete pavement.

RayMan-generated fisheye SVF hemispheres of four urban configurations in the north–south canyon direction (Surya Magna)

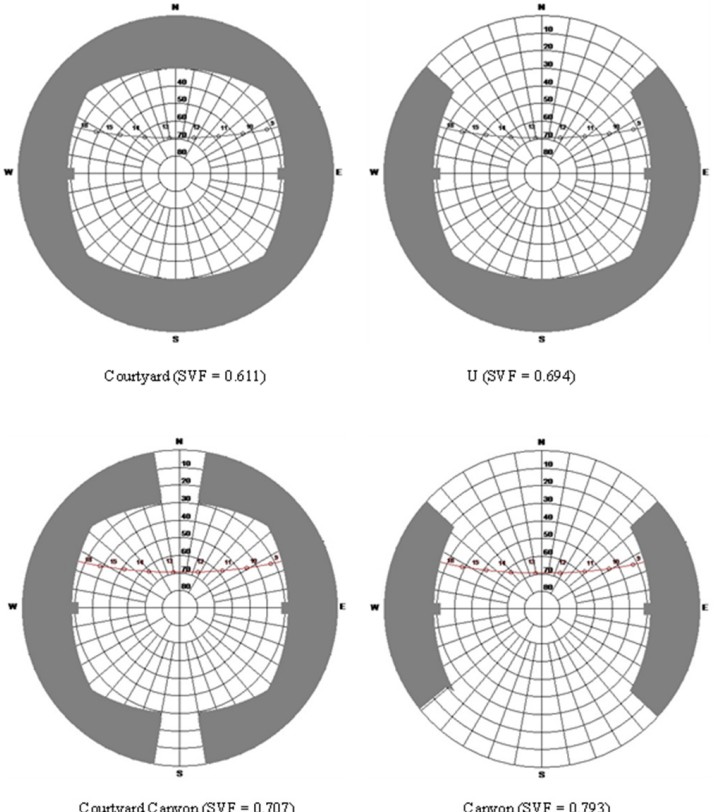

Courtyard (SVF = 0.611)

U (SVF = 0.694)

Courtyard Canyon (SVF = 0.707)

Canyon (SVF = 0.793)

### 3.2. Research Methodology

This study explored a combination of research techniques and mathematical models to evaluate the impact of urban configurations on thermal perceptions at the sites. The following steps were taken to accomplish the research goals:

1. Mapping of the case studies to understand the configurations. The information gathered and mapping of the sites helped us to determine the configurations, including hypothetical and non-hypothetical scenarios. The thermophysical properties of the building blocks and other urban elements modelled for the simulation are presented in Table 3.

2. Field measurements of environmental parameters were carried out to assess the outdoor thermal environmental conditions of the locations. Some of the environmental variables that were measured included air temperature, solar radiation, air velocity, surface temperature, mean radiant temperature, humidity, and others. Existing research has examined the reliability of urban microclimate modelling [101,102]. A similar approach was also adopted in this study.

3. Recent studies [102,103] have reviewed and recommended the need for an empirical simulation approach. Such an approach is also considered in the present study. We used the ENVI-met V3.1 Beta version as the 3D non-hydrostatic microclimate computer simulation for the experimental setup. We selected ENVI-met simulation because it is a reliable tool that suits the objective of our research to investigate both microclimate and thermal comfort variables with different parameters and determine the gaps. ENVI-met not only generates the statistical results of parameters, but also presents 2D and 3D graphics at high resolution. Moreover, ENVI-met simulation has a strong focus on microscale urban space investigation. Some significant studies regarding the reliability and scope of ENVI-met were reviewed [104–107]. We applied the grid cells of x = 210, y = 210, and z = 30, with a nesting grid of 6. The main ENVI-met model domain in this study was set in three dimensions, with two horizontal dimensions of x and y and one vertical dimension of z. In general, ENVI-met V3.1 Beta version offered three sizes of simulation sections: $100 \times 100 \times 300$, $180 \times 180 \times 30$, and $250 \times 250 \times 30$. Based on the consideration of the size of the model domain, simulation duration, and minimising the errors in the microscale simulation, we applied the $250 \times 250 \times 30$ grid setting. However, the analysis of this study focused on the horizontal dimensions of x and y at the pedestrian-height microclimate and outdoor thermal comfort indices. To ensure the proper height of the actual building and the model, a telescoping method was used with a 20% factor that started at 2 m. The model was rotated 5° from the north direction to set the actual direction of the site. The geographical location was set in Kuala Lumpur, Malaysia (3°08′0.5″ N, 101°041′0.36″ E), for the sites.

4. As it is geographically located in a tropical region, Malaysia has constant high-intensity solar radiation throughout the year, causing its significant hot and humid climate characteristics. The solar radiation pattern is different each year, and the highest solar radiation may fall in any of the months throughout the year [108]. Solar radiation is the main source of heat stress in the modification of thermal indices (i.e., microclimate and thermal comfort) in the context of tropical regions. This study was conducted in June—a month with stable global radiation and one of the hottest of the year. The uniform high solar radiation throughout the year is consistent with the uniform high temperature in Malaysia. In this study, 21 June was chosen as the simulation date because it is the longest day of the year for the area north of the equator.

The input parameters used for the ENVI-met simulations were decided according to the requirements of study and previous site observations. However, the climatic input data were derived from the annual average data from Kuala Lumpur's meteorological department and a review of previous studies [109–112]. The data are outlined in Table 4.

**Table 4.** Summary of the input parameters considered for the simulations.

| Variables | Corresponding Values |
|---|---|
| The start date of the simulation | 21 June |
| The start time of the simulation | 06.00.00 |
| Total simulation time (hours) | 24.0 |
| Save model duration (min) | 60.0 |
| Wind speed in (m/s) | 1.4 |
| Wind direction | 225 (Southwest) |
| Roughness length z0 at the reference point | 0.1 |
| Initial temperature atmosphere (K) | 303.15 |
| Specific humidity in 2500 m | 4.0 |
| Relative humidity in 2 m (%) | 83.0 |
| Initial upper-layer temperature (0–20 cm) (K) | 303.15 |
| Relative upper-layer humidity (0–20 cm) (%) | 83.0 |
| Internal temperature of building (K) | 293.0 |
| Heat transmission—walls (W/m$^2$K) | 1.94 |
| Heat transmission—rooves (W/m$^2$K) | 6.0 |
| Albedo walls | 0.3 |
| Albedo roofs | 0.5 |
| Walking speed (m/s) (for PMV) | 0.0 |
| Energy exchange (Col. 2 M/A) (for PMV) | 70.0 |
| Mechanical factor (for PMV) | 0.0 |
| Heat transfer resistance of clothes (for PMV) | 0.5 |

## 4. Results

The simulated data were first calibrated and validated using the measured data from the field measurements carried out in each location of the existing urban configuration (i.e., courtyard canyon). The calibration and validation of the simulated data revealed similar patterns between the measured and simulated results, as shown in Figures 2–7. The validation results from both sites show that both the measured parameters and the simulated data followed a similar pattern—especially data from the SM site, as shown in Figures 5 and 6.

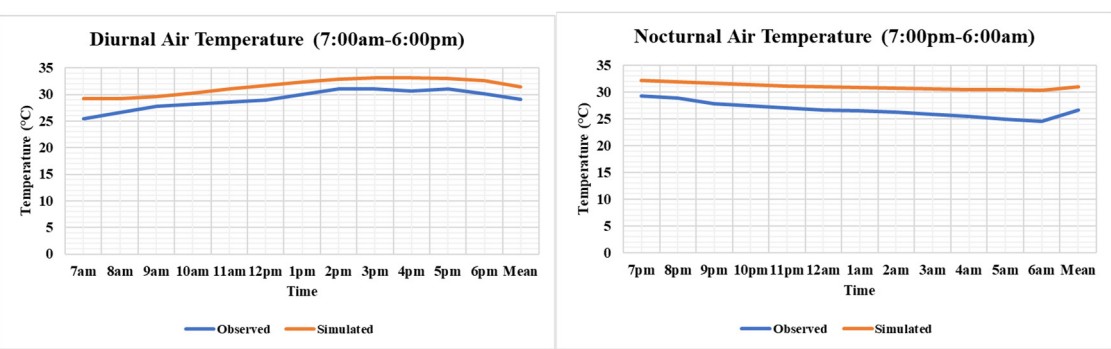

**Figure 2.** Measured and simulated results of diurnal and nocturnal air temperatures at FBTS.

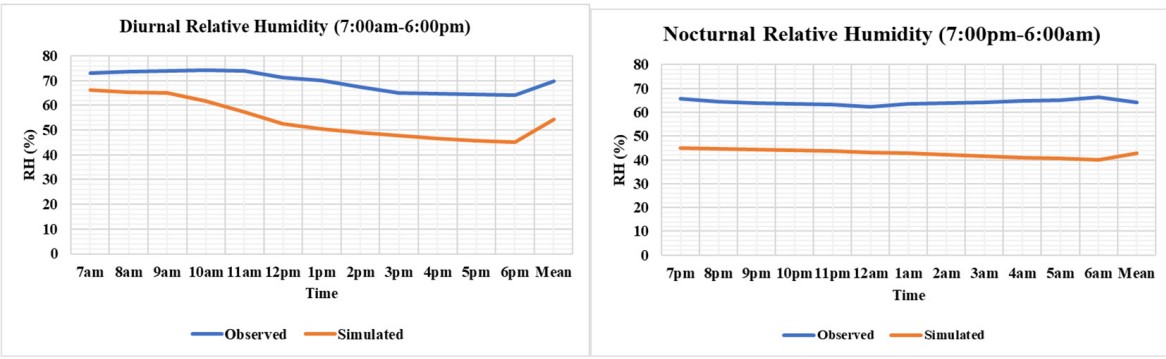

**Figure 3.** Measured and simulated results of diurnal and nocturnal relative humidity at FBTS.

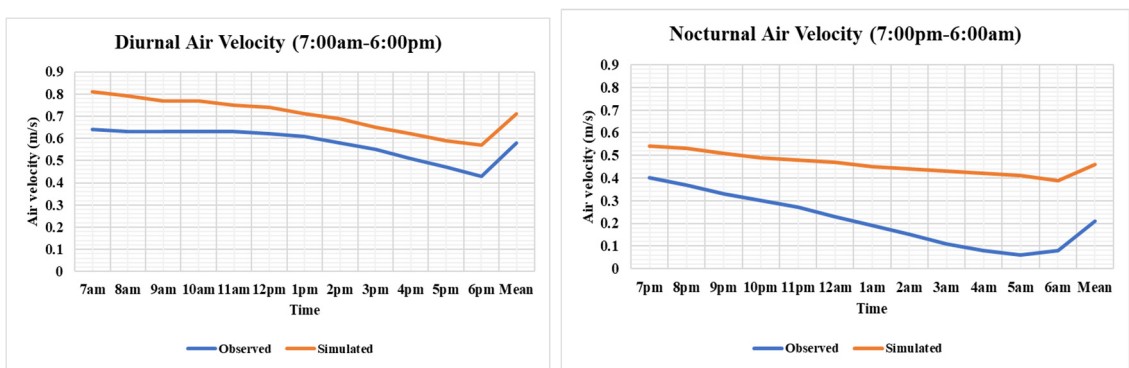

**Figure 4.** Measured and simulated results of diurnal and nocturnal air velocity at FBTS.

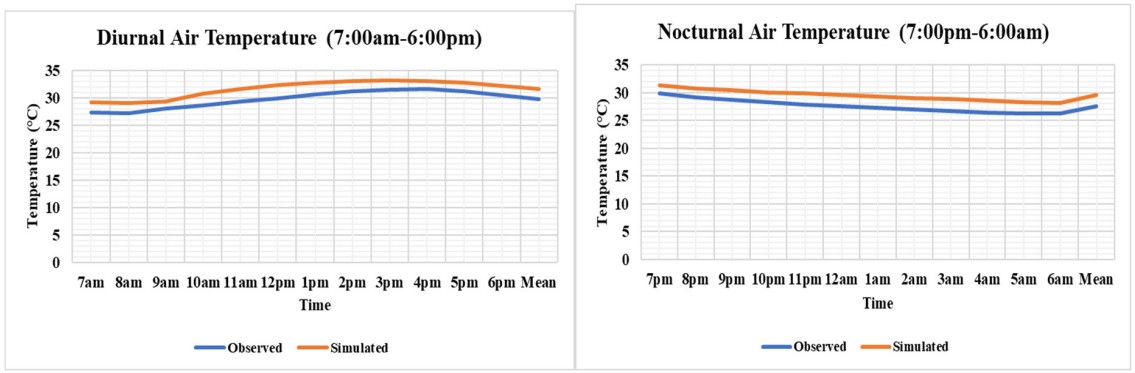

**Figure 5.** Measured and simulated results of diurnal and nocturnal air temperatures at SM.

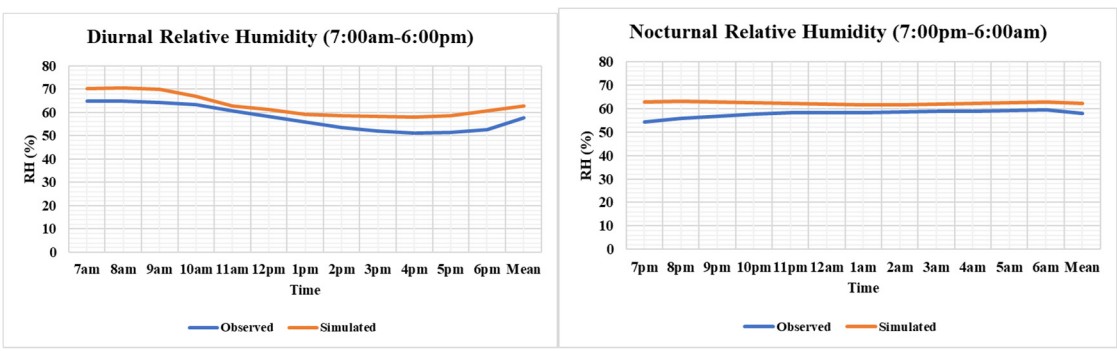

**Figure 6.** Measured and simulated results of diurnal and nocturnal relative humidity at SM.

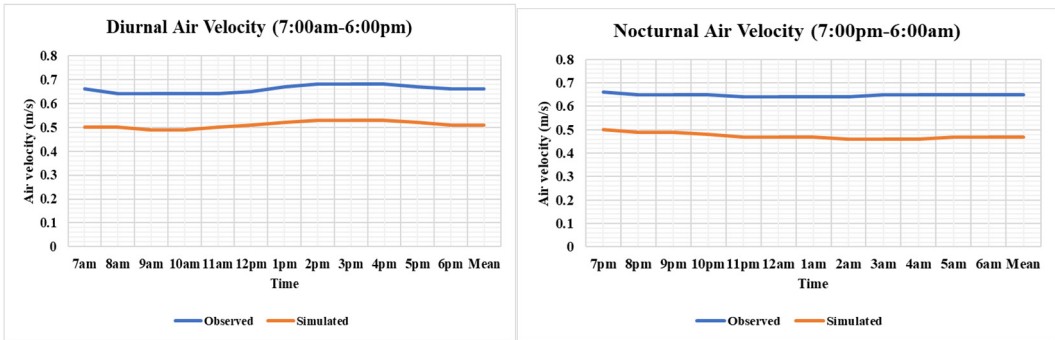

**Figure 7.** Measured and simulated results of diurnal and nocturnal air velocity at SM.

We also considered regression analyses for the validation of the observed and simulated data. The analysis revealed strong correlations between the two sets of data, with a much stronger correlation found at SM compared to FBTS, as shown in Figures 8 and 9. Strong correlation for air velocity was also recorded at the FBTS site ($R^2$ = 0.939, RMSE = 0.035 m/s, $p$ = 0.00) and the SM site (R2 = 0.602, RMSE = 0.015 m/s, $p$ = 0.00). However, the site observation of the Tmrt was not performed due to limited resources in terms of validation from the occupants.

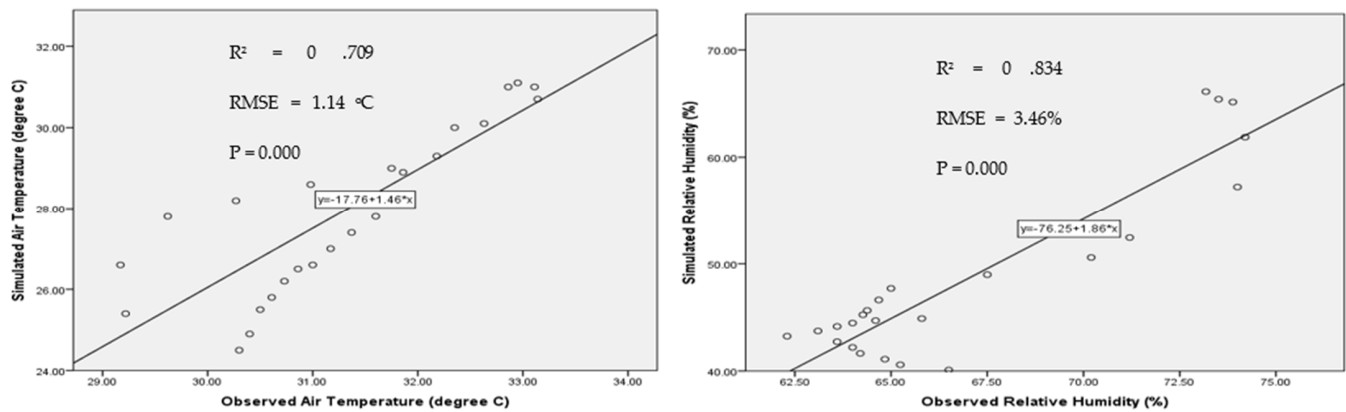

**Figure 8.** Regression of ENVI-met validation between measured and simulated air temperature (**left**) and RH (**right**) at FBTS.

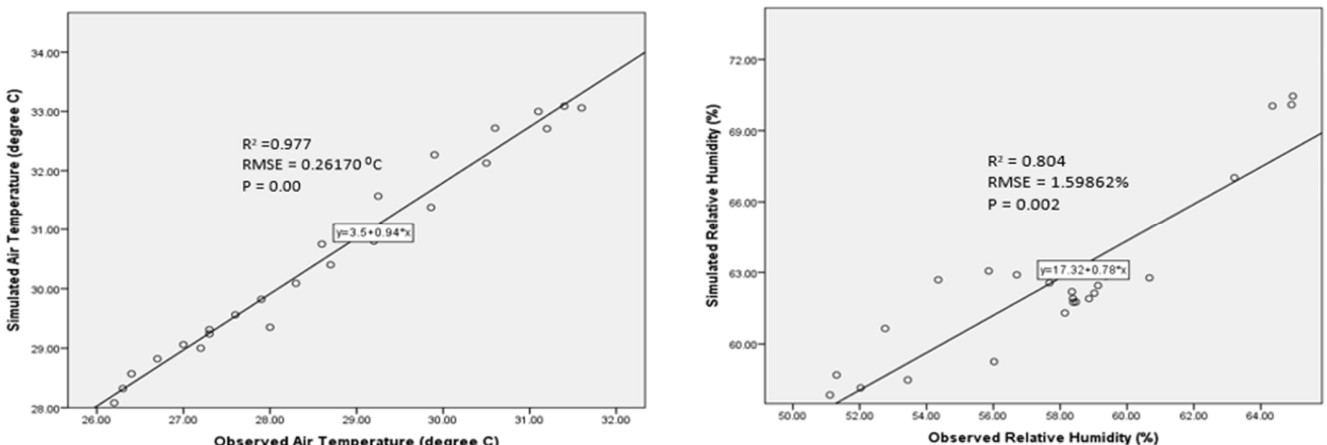

**Figure 9.** The validation between measured and simulated temperature (**left**) and RH (**right**) at SM.

In Case Study 1—Flat Bandar Tasik Selatan (FBTS), the analysis of the surface temperature revealed noticeable nocturnal differences between urban configurations with and without canyons. During the daytime, a different pattern was noted between urban configurations with and without canyons, except at noon. In Case Study 2—Surya Magna (SM), an identical trend but a smaller gap was noted in the nocturnal surface temperature (Figure 10). In the analysis of solar radiation at FBTS, during the day, shortwave radiation was noted.

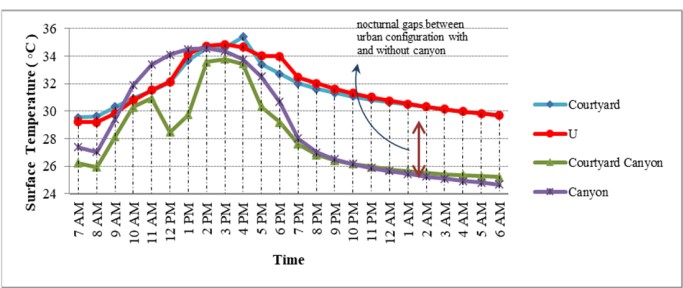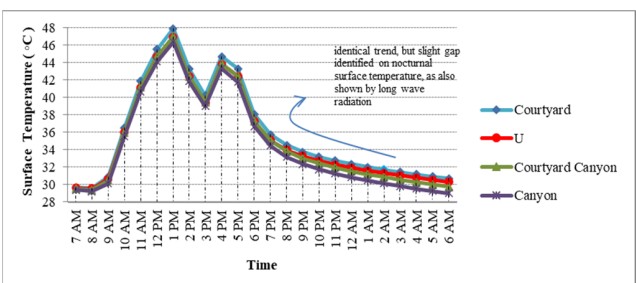

**Figure 10.** Surface temperature vs. time at FBTS (**left**) and SM (**right**).

In terms of the mean radiant temperature (Tmrt) at FBTS, for the diurnal temperature, the variable (Tmrt) varied as a result of variation in vertical obstruction in urban configurations. At SM, a similar trend to solar radiation, shortwave radiation, and longwave radiation was noted, along with minimal gaps in Tmrt among the configurations (Figure 11). Regarding air velocity at FBTS, more gaps between urban geometries with and without canyon features were noted, and the gaps were greater during the day than those observed at night. The analysis of air temperature showed that higher nocturnal air temperatures were noted compared to the diurnal values in the enclosed canyons at FBTS. At SM, enclosed urban geometries of courtyards and courtyard canyons led to higher nocturnal air temperatures compared to urban geometries with canyon characteristics (Figure 12). In terms of relative humidity at FBTS, there was nearly uniform diurnal relative humidity between urban geometries when the solar radiation was maximal. At SM, wider gaps between urban geometries with canyon features and enclosed urban geometries were noted during the day and night than were observed at FBTS (Figure 13).

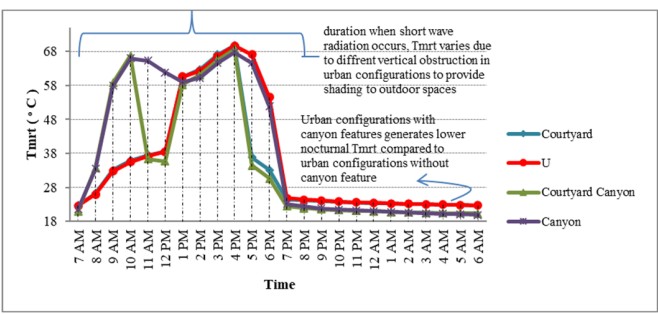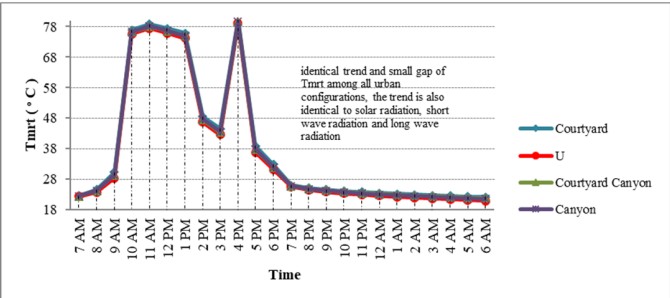

**Figure 11.** Mean radiant temperature vs. time at FBTS (**left**) and SM (**right**).

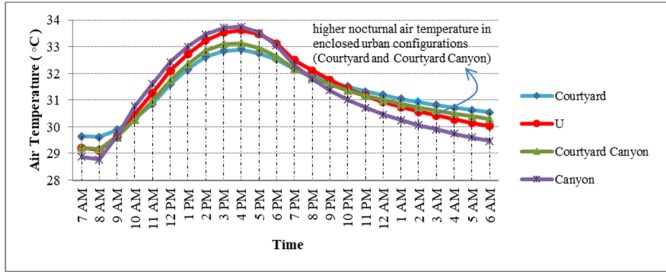 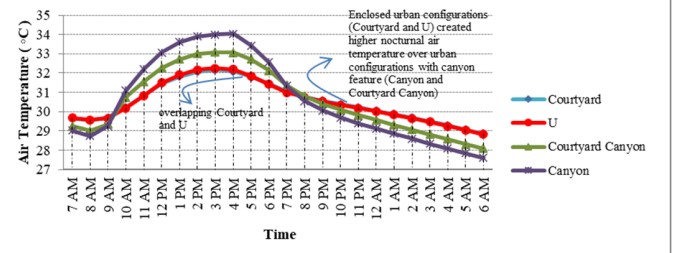

**Figure 12.** Air temperature vs. time at FBTS (**left**) and SM (**right**).

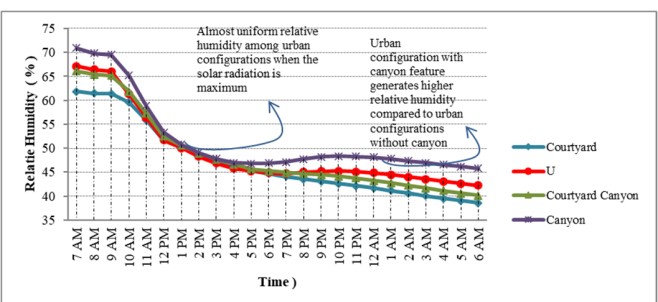 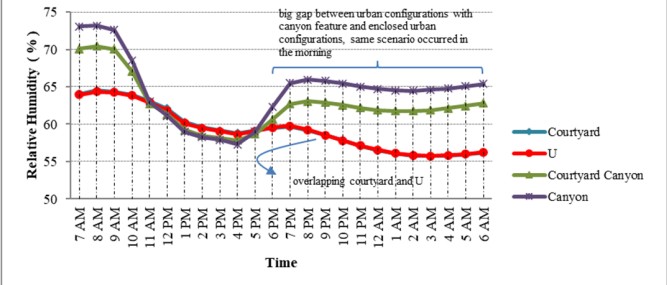

**Figure 13.** Relative humidity vs. time at FBTS (**left**) and SM (**right**).

We also analysed outdoor thermal indices (i.e., WBGT, SET, and UTCI) and considered the averages of the variables in the case studies. The aim was to understand the impact of urban configurations on outdoor thermal indices. The analysis revealed that across the case studies, higher mean values of air temperature, solar radiation, radiant temperature, surface temperature, and operative temperature were reported during the day than during the night in the different models. The analysis showed lower thermal indices at SM than at FBTS (Table 5).

**Table 5.** Mean diurnal and nocturnal values of environmental variables and thermal indices for various urban configurations in the case studies.

| Urban Configurations/Variables | | Mean Air Temp. (°C) | Mean Solar Radiation (W/m²) | Mean Radiant Temp. (°C) | Mean Surf. Temp. (°C) | Mean Oper. Temp. (°C) | Mean RH (%) | Mean Air Vel. (m/s) | Mean WBGT (°C) | Mean SET (°C) | Mean UTCI (°C) |
|---|---|---|---|---|---|---|---|---|---|---|---|
| | | **Diurnal** | | | | | | | | | |
| Courtyard (SVF: 0.275) | | 31.48 | 812.70 | 43.69 | 32.37 | 38.35 | 52.83 | 0.06 | 26.34 | 38.60 | 35.40 |
| U (SVF: 0.309) | Case Study 1—FBTS | 31.79 | 905.64 | 47.76 | 32.43 | 42.10 | 54.16 | 0.03 | 26.77 | 42.90 | 36.81 |
| Courtyard canyon (SVF: 0.438) | | 31.50 | 935.86 | 47.59 | 30.01 | 35.89 | 54.41 | 0.71 | 26.54 | 33.80 | 36.52 |
| Canyon (SVF: 0.676) | | 31.88 | 1204.90 | 56.16 | 31.96 | 39.04 | 56.31 | 0.57 | 27.15 | 37.70 | 31.22 |
| Courtyard (SVF: 0.611) | | 31.07 | 998.49 | 52.66 | 39.31 | 39.64 | 61.53 | 0.23 | 27.08 | 40.80 | 37.33 |
| U (SVF: 0.694) | Case Study 2—SM | 31.10 | 2077.23 | 51.86 | 38.66 | 38.14 | 61.41 | 0.38 | 27.09 | 38.20 | 37.35 |
| Courtyard canyon (SVF: 0.707) | | 31.57 | 3075.72 | 51.93 | 38.66 | 37.82 | 62.90 | 0.51 | 27.67 | 37.70 | 37.96 |
| Canyon (SVF: 0.793) | | 32.08 | 1005.82 | 51.19 | 38.20 | 37.35 | 63.76 | 0.69 | 28.18 | 36.90 | 38.57 |

**Table 5.** *Cont.*

| Urban configurations/variables | | Mean Air Temp. (°C) | Mean Solar Radiation (W/m²) | Mean Radiant Temp. (°C) | Mean Surf. Temp. (°C) | Mean Oper. Temp. (°C) | Mean RH (%) | Mean Air Vel. (m/s) | Mean WBGT (°C) | Mean SET (°C) | Mean UTCI (°C) |
|---|---|---|---|---|---|---|---|---|---|---|---|
| | | | | | Nocturnal | | | | | | |
| Courtyard (SVF: 0.275) | Case Study 1-FBTS | 31.21 | 483.06 | 23.41 | 30.66 | 27.31 | 41.33 | 0.10 | 24.53 | 26.80 | 29.01 |
| U (SVF: 0.309) | | 31.01 | 483.99 | 23.50 | 30.80 | 26.92 | 44.16 | 0.07 | 24.78 | 26.60 | 29.08 |
| Courtyard canyon (SVF: 0.438) | | 31.05 | 453.67 | 21.06 | 25.93 | 27.87 | 42.76 | 0.46 | 24.61 | 25.20 | 28.36 |
| Canyon (SVF: 0.676) | | 30.56 | 452.78 | 21.08 | 25.78 | 27.67 | 47.36 | 0.52 | 24.83 | 25.00 | 28.28 |
| Courtyard (SVF: 0.611) | Case Study 2-SM | 29.94 | 495.10 | 23.67 | 32.53 | 27.01 | 57.02 | 0.13 | 25.43 | 27.20 | 29.24 |
| U (SVF: 0.694) | | 29.91 | 492.08 | 23.34 | 32.06 | 27.27 | 57.07 | 0.22 | 25.41 | 26.40 | 29.13 |
| Courtyard canyon (SVF: 0.707) | | 29.52 | 987.18 | 23.17 | 31.71 | 27.78 | 62.34 | 0.47 | 25.74 | 25.80 | 29.21 |
| Canyon (SVF: 0.793) | | 29.12 | 485.40 | 22.71 | 31.02 | 27.41 | 65.10 | 0.76 | 25.69 | 24.70 | 28.96 |

## 5. Discussion

The results at FBTS showed high intensity in terms of surface temperature in the courtyard. The enclosed urban geometries enhanced short- and longwave radiation. In the U courtyard in the same case study, the highest noted intensity was directed to the west. Moreover, the east side of the model was blocked and tended to trap the longwave radiation. In the courtyard canyon model at FBTS, the lowest surface temperatures were reported. As a result, small canyons tended to reduce the trapped longwave radiation. For the canyon, the features decreased the longwave radiation. In Case Study 2 (SM), the highest surface temperatures were noted in the courtyard model. The longwave radiation was trapped due to the heat that was blocked from the different orientations. In the U model, high surface temperatures were radiated and partly blocked from ventilation. In the courtyard canyon, a small channel effect was observed in this case. In the canyon model, the features of the model decreased the trapped longwave radiation effectively.

In terms of solar radiation at FBTS, the lowest radiation was noted in a fully enclosed area with shading effects. In the U model, slightly higher radiation was noted due to longwave radiation. On the one hand, in the courtyard canyon, the east and west orientations were exposed to solar radiation. On the other hand, in the canyon model, the highest solar radiation was observed from the east and west directions. At SM, low solar radiation was observed in the courtyard, and the urban blocks were shaded from solar exposure in all orientations. The U model was exposed to indirect solar exposure from the north, and the radiation was considerably influenced by the high June solar radiation in the location. In the courtyard canyon, the model was enclosed but moderately exposed to northern solar radiation. In the canyon, the lowest solar radiation was noted in the model with urban blocks shading the open space from eastern and western solar exposure.

At FBTS, the lowest mean radiant temperature (Tmrt) was found on the four-sided urban blocks of the courtyard, which provided shading from east–west solar radiation exposure. In the U courtyard, a high Tmrt was noted; the model was highly exposed to western solar radiation, and no wind channel effect was provided. In the courtyard canyon, low mean radiant temperatures were observed. Both shading and ventilation were provided. In the canyon, the highest Tmrt was predicted in the model. Additionally, both the east and west sides were prone to maximum solar radiation exposure. Across the models at SM, the highest Tmrt was noted in the courtyard model, and the enclosed open spaces amplified longwave solar radiation. In the U courtyard, the model indicated high Tmrt; it was exposed to northern solar radiation, and no wind channel effect was observed. The result of the courtyard canyon model showed low-Tmrt canyon features, while the lowest Tmrt was predicted because of the canyon features.

In terms of air temperature at FBTS, in the courtyard model, high nocturnal air temperatures were noted in the areas with the smallest SVF. As a result, the enclosed

outdoor spaces generated shading effects and trapped the longwave radiation, and no channel effect was noted. In the U model, the highest air temperature was predicted due to high solar radiation from the west. In the courtyard canyon, the lowest air temperature was noted, encouraging the channel effect. Likewise, in the canyon, the model revealed the highest SVF, with a channel effect and high shortwave radiation. At SM, the lowest temperatures in enclosed outdoor spaces were noted in the courtyard model. Similar results were obtained in the U and courtyard canyon models. In the canyon model, the highest temperature was noted, with the highest SVF and high northern radiation exposure.

We also examined the possible air temperature and mean radiant temperature (Tmrt) ranges at 0 m above the ground level at 4 p.m. in various models at FBTS and SM (Figures 14–17). Across the models, the investigated urban configurations are highlighted in black. At FBTS, a lower range of temperatures was noted in the courtyard and courtyard canyon models than the values predicted in the U and canyon models. The enclosed configurations of the courtyard and courtyard models may be a contributing factor to this outcome. In the second case study (SM), lower temperatures were predicted in the courtyard and U models than the temperatures observed in the courtyard canyon and canyon models. In both case studies, increasing temperatures (above 35.20 °C) were also predicted in many of the models—especially in the canyons. In terms of mean radiant temperature (Tmrt), the mean values in the FBTS models were either under 36.91 °C or within the range of 36.91–42.38 °C, except in the canyon model. At SM, the Tmrt values across the models exceeded the range observed in the models at FBTS. These findings show that occupants in the courtyard models are less susceptible to increasing Tmrt than occupants in other models.

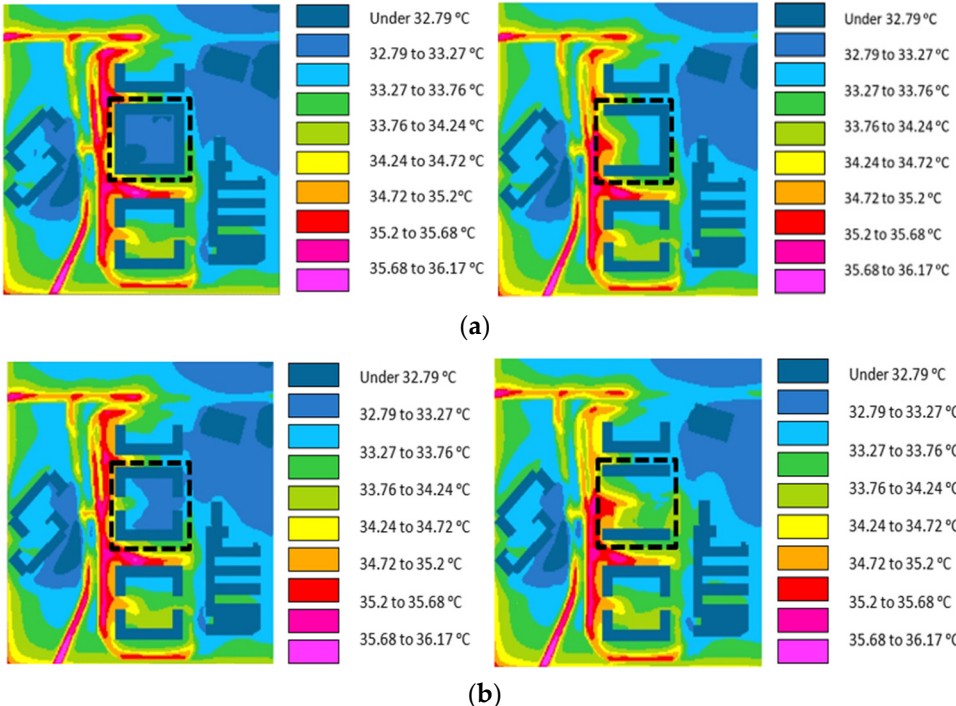

**Figure 14.** (**a**) Predicted air temperature bands at 0 m above the ground level at 4 p.m. in the FBTS models (Left—courtyard; Right—U model). (**b**) Predicted air temperature bands at 0 m above the ground level at 4 p.m. in the FBTS models (Left—courtyard canyon; Right—canyon).

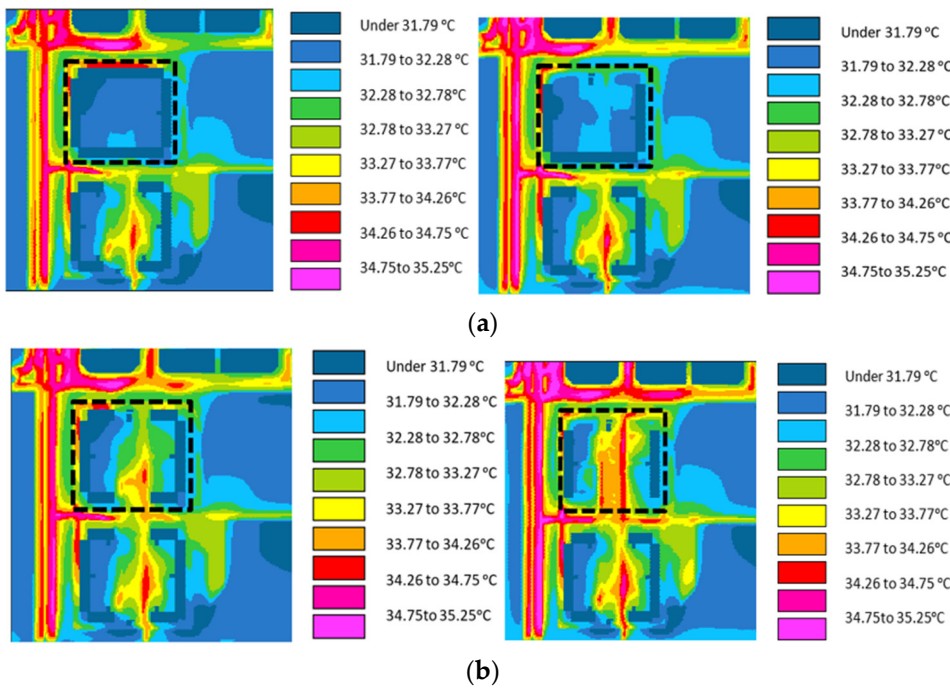

(**a**)

(**b**)

**Figure 15.** (**a**) Predicted air temperature bands at 0 m above the ground level at 4 p.m. in the SM models (Left—courtyard; Right—U model). (**b**) Predicted air temperature bands at 0 m above the ground level at 4 p.m. in the SM models (Left—courtyard canyon; Right—canyon).

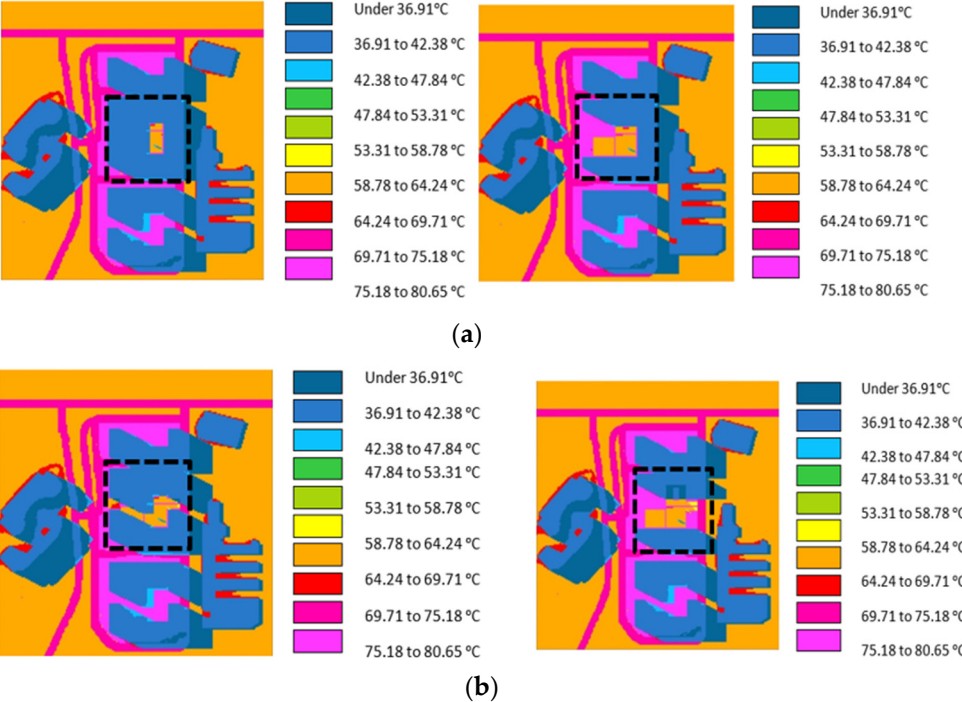

(**a**)

(**b**)

**Figure 16.** (**a**) Predicted mean radiant temperature bands at 0 m above the ground level at 4 p.m. in the FBTS models (Left—courtyard; Right—U model). (**b**) Predicted mean radiant temperature bands at 0 m above the ground level at 4 p.m. in the FBTS models (Left—courtyard canyon; Right—canyon).

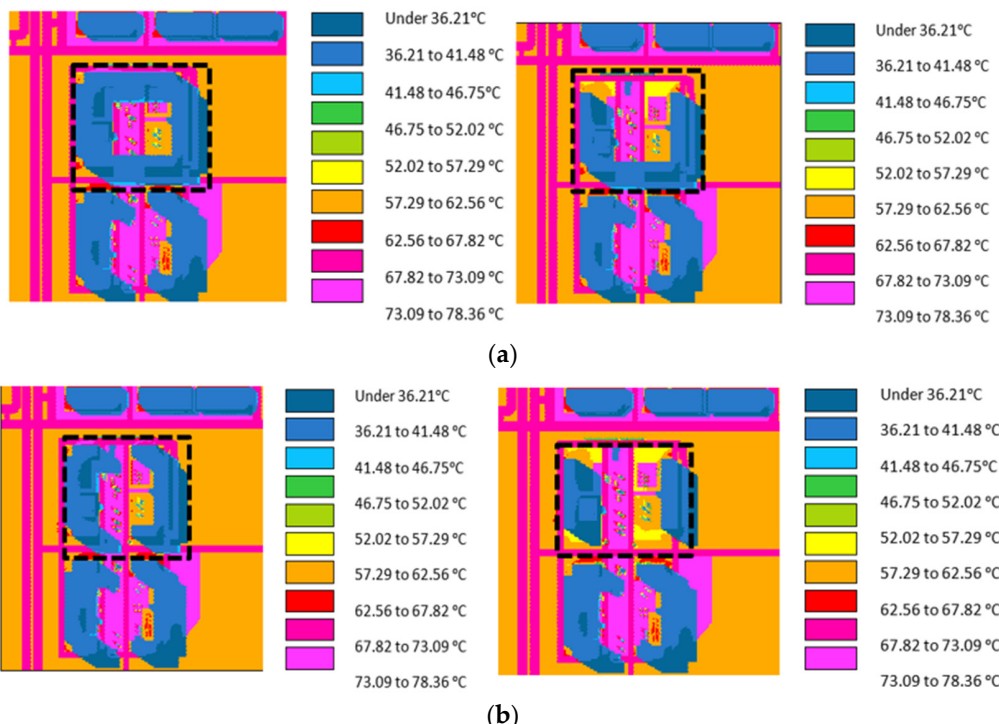

**Figure 17.** (**a**) Predicted mean radiant temperature bands at 0 m above the ground level at 4 p.m. in the SM models (Left—courtyard; Right—U model). (**b**) Predicted mean radiant temperature bands at 0 m above the ground level at 4 p.m. in the SM models (Left—courtyard canyon; Right—canyon).

Concerning air velocity at FBTS, low air velocity as a result of enclosed outdoor spaces with no channel effect was noted in the courtyard model. The lowest wind speed was observed in the U model, due to only one side being open for air flow. The air velocity in the U model was still considerably lower than the value obtained in the courtyard model. In the courtyard canyon, the channel effect was well generated because of the small canyon feature. Likewise, in the canyon model, the channel effect also occurred. At SM, the lowest velocity was noted in the courtyard model, due to enclosed outdoor spaces that obstructed air flow. In the U model, low wind speed was also noted, as observed in the model at FBTS, due to only one side being open for ventilation access. In the courtyard canyon, the channel effect occurred because of the small canyon feature. In the canyon model, the channel effect was well generated and produced the best ventilation scenario to reduce the heat and improve thermal comfort.

We also considered different statistical tests to establish relationships between the variables. At FBTS, significant correlation was reported between air temperature and solar radiation in the courtyard (F = 12.495, $p$ = 0.002, R2 = 0.362), U (F = 33.799, $p$ = 0.000, R2 = 0.606), and canyon models ( F = 19.915, $p$ = 0.000, R2 = 0.475). F represents the variance of the group means. Significant correlation was not reported between the variables in courtyard canyon model at FBTS (F = 4.019, $p$ = 0.057, R2 = 0.154). In the courtyard canyon model, a change in air temperature did not significantly impact solar radiation. At SM, significant correlation was not observed between the variables in the courtyard model (F = 8.854, $p$ = 0.007, R2 = 0.287). In the U (F = 16.093, $p$ = 0.001, R2 = 0.422), courtyard canyon (F = 21.253, $p$ = 0.000, R2 = 0.491), and canyon models (F = 17.195, $p$ = 0.000, R2 = 0.439), significant correlation was reported between air temperature and solar radiation across the three models. The research outcomes showed that air temperature was directly proportional to solar radiation in the U, courtyard canyon, and canyon models. There was a weak association between the variables in the courtyard model at SM.

Significant correlation was not reported between air temperature and air velocity in the courtyard (F = 4.555, $p$ = 0.507, R2 = 0.020), U (F = 2.456, $p$ = 0.131, R2 = 0.100), and

courtyard canyon models (F = 0.089, *p* = 0.768, R2 = 0.004) at FBTS. However, significant correlation was noted between the variables in the canyon model (F = 261.694, *p* = 0.000, R2 = 0.922) at FBTS. Meanwhile, at SM, significant correlation was noted between the parameters in the courtyard (F = 11.984, *p* = 0.002, R2 = 0.353), U (F = 9.516, *p* = 0.005, R2 = 0.302), and courtyard canyon models (F = 86.436, *p* = 0.000, R2 = 0.797). Similar to the results obtained at FBTS, significant correlation was not reported between the variables in the canyon model (F = 2.207, *p* = 0.152, R2 = 0.091) at SM. The results at SM showed that air velocity had a significant impact on air temperature, which could affect the thermal comfort of occupants in the courtyard, U, and courtyard canyon models, and vice versa at FBTS. In the canyon model at FBTS, there was a link between the variables, while no association was noted between the two parameters in the canyon model at SM.

We also considered the overall mean values of the variables and thermal indices at FBTS and SM (Table 6). Our findings showed that higher mean air temperatures were predicted in the models including canyons at FBTS than at SM. The mean values of solar radiation, radiant temperature, surface temperature, and RH at SM were predicted to be higher than the values computed for FBTS. The orientation (north–south) and geometries of the urban blocks in the models at SM—especially the U, courtyard canyon, and canyon models—may have contributed to these outcomes. In terms of thermal indices, those at FBTS ranged from 25.44 to 34.75 °C. At SM, the thermal indices varied from 26.23 to 33.77 °C. Higher mean values of WBGT and UTCI were reported at SM than at FBTS. In terms of the mean values of SET, the highest value was noted at FBTS. The higher mean values of air velocity and RH predicted at SM may be contributing factors to the elevated thermal indices noted at SM compared to FBTS. The investigation showed that urban configurations, in addition to design (e.g., orientation, shading effect, etc.) and environmental parameters (e.g., air velocity, RH, Tmrt, etc.), can influence thermal indices, and that temperatures experienced by outdoor occupants can render them prone to thermal stress within the thermal environment.

**Table 6.** Overall mean values of environmental variables and thermal indices for various urban configurations in the case studies.

| Urban Configurations/Variables | | Mean Air Temp. (°C) | Mean Solar Radiation (W/m²) | Mean Radiant Temp. (°C) | Mean Surf. Temp. (°C) | Mean Oper. Temp. (°C) | Mean RH (%) | Mean air Vel. (m/s) | Mean WBGT (°C) | Mean SET (°C) | Mean UTCI (°C) |
|---|---|---|---|---|---|---|---|---|---|---|---|
| Courtyard (SVF: 0.275) | | 31.34 | 647.88 | 33.60 | 31.51 | 32.83 | 47.08 | 0.08 | 25.44 | 32.55 | 32.21 |
| U (SVF: 0.309) | Case Study 1—FBTS | 31.40 | 694.82 | 35.60 | 31.62 | 34.51 | 49.16 | 0.05 | 25.78 | 34.75 | 32.95 |
| Courtyard canyon (SVF: 0.438) | | 31.28 | 694.76 | 34.30 | 27.97 | 31.88 | 48.60 | 0.58 | 25.58 | 29.50 | 32.44 |
| Canyon (SVF: 0.676) | | 31.22 | 828.84 | 38.60 | 28.87 | 33.36 | 51.83 | 0.54 | 25.99 | 31.35 | 29.75 |
| Courtyard (SVF: 0.611) | | 30.50 | 746.79 | 38.16 | 35.92 | 33.33 | 59.27 | 0.18 | 26.23 | 34.00 | 33.29 |
| U (SVF: 0.694) | Case Study 2—SM | 30.50 | 1284.66 | 37.60 | 35.36 | 32.71 | 59.24 | 0.30 | 26.25 | 32.30 | 33.24 |
| Courtyard canyon (SVF: 0.707) | | 30.55 | 2031.45 | 37.55 | 35.18 | 32.80 | 62.62 | 0.49 | 26.71 | 31.75 | 33.59 |
| Canyon (SVF: 0.793) | | 30.60 | 745.61 | 36.95 | 34.61 | 32.38 | 64.43 | 0.72 | 26.94 | 30.80 | 33.77 |

Additionally, we compared the findings obtained in this study with previous research on assessments of urban configurations in outdoor spaces [6,7,40,68] and thermal indices in outdoor spaces [7,89]. For instance, when comparing the simulated air temperatures from the current study with an existing model [68], our findings revealed variations that ranged from 0.3 °C to 3.0 °C between diurnal and nocturnal temperatures (Table 7). In most cases, an increase in air temperature was noted across various models. The findings of the comparison are presented in the table below. Comparing the mean thermal indices obtained in this study with existing research [7,89], higher thermal indices were reported

in this study than the values obtained in the previous investigations. Our findings showed higher thermal indices in the study locations than the values obtained in other regions.

**Table 7.** Comparison of the simulated air temperatures between the present study and previous research [40,68].

| Urban Configurations/Variables | | Estimated $dT_{max} = 15.3 - 13.9$ (* SVF)—°C | Diurnal Temp. (°C) | Nocturnal Temp. (°C) | Difference between Diurnal and Nocturnal Temp. (°C) | Notes |
|---|---|---|---|---|---|---|
| Courtyard (SVF: 0.275) | | 11.5 | 31.5 | 31.2 | 0.3 | Previous research showed the smallest SVF compared to the current study. Furthermore, the smallest change between diurnal and nocturnal temperatures was observed |
| U (SVF: 0.309) | | 11.0 | 31.8 | 31.0 | 0.8 | A decrease in the nocturnal temperature should be lower than the value obtained for the courtyard canyon, which has a larger SVF. The results were consistent with those of previous research |
| Courtyard canyon (SVF: 0.438) | Case Study 1—FBTS | 9.2 | 31.5 | 31.1 | 0.5 | The diurnal and nocturnal temperatures were within the same range as the values obtained for the courtyard. The difference between the diurnal and nocturnal temperature was 0.2 °C higher than the value obtained for the courtyard. The findings were consistent with those of the previous model |
| Canyon (SVF: 0.676) | | 5.9 | 31.9 | 30.6 | 1.3 | In this model, the largest value of SVF was noted. Likewise, the largest difference between the diurnal and nocturnal temperatures was also predicted |
| Courtyard (SVF: 0.611) | | 6.8 | 31.1 | 29.9 | 1.2 | In this model, the smallest difference between the diurnal and nocturnal temperatures and the smallest SVF were noted |
| U (SVF: 0.694) | | 5.6 | 31.1 | 29.9 | 1.2 | Similar results to those obtained in the courtyard model were obtained in this model, with a higher SVF. The results were consistent with previous research |
| Courtyard canyon (SVF: 0.707) | Case Study 2—SM | 5.5 | 31.6 | 29.5 | 2.1 | An increase (from 1.2 to 2.1 °C) in the difference between the diurnal and nocturnal temperatures was noted when compared to the values noted in the courtyard and U models. The result was also consistent with the previous model analysed in this study |
| Canyon (SVF: 0.793) | | 4.3 | 32.1 | 29.1 | 3.0 | The model revealed the highest value for SVF. Likewise, the highest difference between the diurnal and nocturnal temperatures was also noted, as observed in the previous model |

Overall, our results present the impact of urban configurations on urban microclimate and thermal comfort, as well as the suggested range of thermal indices. The finding of this study are consistent with those of previous studies [18,29,69,98,104], showing that urban configuration (specifically the canyon features in the urban configurations) significantly impacts the modification of microclimate and thermal comfort variables (i.e., air temperature, relative humidity, air velocity, and mean radiant temperature). However, this study adds strong findings on the impact of the urban configuration on the thermal indices in the context of outdoor environments in tropical, hot and humid locations ranging between 25.44 and 34.75 °C.

In terms of the research limitations, this study assessed two sites in the study location and could not assess additional sites in different regions within the country. Future work should explore and assess more sites across different regions across the country. Moreover, future work should consider outdoor comfort using these models in different seasons. However, the present study considered four different models from different orientations and carried out thorough analyses of these models. The present work addresses the research questions and provides original contributions to the field by examining outdoor thermal comfort in various urban configurations. Our findings also highlight the impact of urban configurations on thermal perceptions in the study location.

## 6. Conclusions

This study assessed and discussed the impact of urban configurations on thermal perceptions in Kuala Lumpur, Malaysia. Based on the outcomes of the investigation, the study presents the following conclusions:

(1) Do urban configurations influence outdoor occupants' comfort in the study locations?

- Our research revealed that urban geometries influence occupants' comfort in the thermal environment.
- We noted that lower air temperatures, mean radiant temperatures, and surface temperatures were predicted in the courtyard canyon and canyon models than the values obtained in the courtyard and U models—especially at FBTS.
- At SM, the mean values of mean radiant temperatures were lower in the courtyard canyon and canyon models than the values predicted in the other models.
- The highest mean values of solar radiation were noted in the canyon models.
- At both locations, the mean values of air velocity and RH were higher in the courtyard canyon and canyon models than the values obtained in the remaining models.
- The geometries, orientations, and SVFs of the courtyard canyon and canyon models appear to be the contributing factors influencing the outcomes obtained in these models.

(2) Do urban configurations also have significant impacts on thermal perceptions in the study areas?

- Our findings showed that air velocity has a minimal impact on WBGT, because higher values of this variable predicted in the canyon models did not significantly reduce the WBGT.
- An increase in RH tended to significantly increase the values of WBGT across the different models at FBTS and SM—especially in the canyon models.
- We also found that an increase in SVF can influence the thermal indices—especially WBGT.
- Our findings showed that urban configurations, in addition to other parameters, can have a significant impact on thermal perceptions in urban areas. Concerning the implications of the study, this investigation revealed the following:
- Continuous exposure of urban areas in tropical regions to high solar radiation can impact the thermal comfort of people. Therefore, appropriate design inter-

ventions can reduce the impact of high solar radiation and other variables in outdoor spaces.

- Our findings also indicate that the outcomes can provide insight on how to assess people's vulnerability to thermal stress in outdoor spaces.

The findings of the present study confirm that Oke's model [68] is applicable to tropical regions. However, this study showed that the intensity of the outdoor thermal stress does depends not only on the Height-to-Width aspect ratios and Sky View Factors (SVFs), but also on the geometry of the urban configurations, including the direction of the Sun's path.

In terms of the applications of this study, the research outcomes can help designers and other professionals to understand various interventions to enhance occupants' comfort in outdoor spaces. Finally, the research outcomes can help policymakers to understand the thresholds for thermal indices for various periods.

**Author Contributions:** Conceptualisation, L.Y. and T.O.A.; methodology, L.Y.; software, L.Y.; validation, L.Y.; formal analysis, L.Y. and T.O.A.; investigation, L.Y.; resources, L.Y.; data curation, L.Y., T.O.A. and O.G.A.; writing—original draft preparation, L.Y. and T.O.A.; writing—review and editing, L.Y., T.O.A. and O.G.A.; visualisation, L.Y. and T.O.A.; supervision, L.Y.; project administration, L.Y. and O.G.A.; funding acquisition, L.Y. All authors have read and agreed to the published version of the manuscript.

**Funding:** This research was funded by Universitas Indonesia. The research team would also like to acknowledge the University of Utah for the Institutional Open Access Program (IOAP) participant's support.

**Data Availability Statement:** Not applicable.

**Conflicts of Interest:** The authors declare no conflict of interest.

## Abbreviations

| | |
|---|---|
| Height-to-Width | H/W |
| Wet-Bulb Globe Temperature | WBGT |
| Universal Thermal Climate Index | UTCI |
| Standard Effective Temperature | SET |
| Urban Heat Island | UHI |
| Mean radiant temperature | Tmrt |
| Sky View Factor | SVF |
| Relative Humidity | RH |

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
