# Peer review of "The Impacts of Urban Configurations on Outdoor Thermal Perceptions: Case Studies of Flat Bandar Tasik Selatan and Surya Magna in Kuala Lumpur"

_buildings, doi:10.3390/buildings12101684_

Round 1

Reviewer 1 Report

The Novelty of this paper is insufficient, so far there are a large number of similar studies. The reason to choose the study site is not clear and the structure of the paper is also confusing.

1.     The correlation between simulation and the site survey study is weak.

2. The difference between the selected thermal indices are not clear enough, the applicability and scope of the thermal indices are not involved, and the conclusion of the article does not compare the applicability of the thermal indices.

3. In the part of literature review on the thermal index, UTCI is the most commonly used thermal index to evaluate outdoor thermal comfort. There are few descriptions in this paper, and relevant information needs to be supplemented, such as the definition of UTCI, specific cases of application, calculation formula, and software used for calculation.

4. The literature review in table 1 is not detailed enough, research sites, simulation methods and thermal index in use need to be provided.

5. The detailed description of specific study site should be provided such as the current satellite map to illustrate, which will be more conducive to readers to understand the different situations of urban configurations.

6. There is a need to supplement the reasons for choosing to use the ENVI-met model, as well as the case or literature on relevant studies.

7. The calculation method of Tmrt(mean radiant temperature)needs to be supplemented.

8. The recording methods and instruments for on-site meteorological data in table 3 require additional details.

9. In Results, with respect to validation of observed and simulated data, the author only analyzed air temperature and humidity when the strong correlation between the two sets of data is revealed ? Why are other meteorological factors ( wind speed and Tmrt ) not verified ?

Details :

1. The word Courtyard in the first line of the Diurnal section in table 4 is spelled incorrectly as Couyard.

2. The fifth line of the fifth paragraph of the Introduction should be followed by ' : ' rather than ' - '.

Author Response

26th September 2022

Guest Editors

Buildings Journal

Dear Editors,

The authors would like to thank the reviewers for the careful review and their comments which have helped to strengthen the quality of the manuscript.

The manuscript entitled ‘The impacts of urban configurations on outdoor thermal perceptions: Case studies of Flat Bandar Tasik Selatan and Surya Magna in Kuala Lumpur’ has been revised accordingly and changes are highlighted in red. Please find attached our response report to the Reviewers.

Thank you, and we look forward to receiving your reply.

Sincerely

Dr. Lin Yola

Senior Lecturer,

School of Strategic and Global Studies, Universitas Indonesia, Jakarta, Indonesia

[email protected]

Reviewer 2 Report

In this paper, the authors investigate the effects of urban layouts on thermal perceptions in Malaysia. The research is critical for comprehending urban design's effects on thermal sensations outdoors. The investigation maps sites to comprehend the configurations to achieve their aims. On-site observations were conducted to evaluate the environmental conditions of the sites. The potential impacts of the locations were modelled and simulated using ENVI-met. The results demonstrated that urban layouts, other design elements (such as orientation towards the sun path), and environmental conditions might affect comfort conditions and conceptions in hot and humid climatic zones. According to the authors' research, the necessity of planning alternative urban designs in hot and humid regions is rooted in the high levels of solar radiation and the desire for a more comfortable thermal environment for outdoor activities.

Thanks for this interesting topic.

Please revise the paper according to these minor comments:

1-    The abstract should clearly identify the key findings of your research.

2-    Please clarify your research novelty compared to previous research.

3-    It would be best if you enhanced the literature review section.

4-    Figures need to be resized and added in a higher-quality format.

5-    The model validation procedures should be clarified as they are unclear.

6-    The manuscript's language should be revised. Several grammatical errors occur.

7-    Also, avoid blending British and American English.

8-    The conclusion section is too long. Please state your findings concisely. You may use bullet points.

Author Response

(The authors gave the same response as above.)

Reviewer 3 Report

The manuscript titled “The impacts of urban configurations on outdoor thermal perceptions: Case studies of Flat Bandar Tasik Selatan and Surya Magna in Kuala Lumpur” investigated the impacts of urban configurations on thermal perceptions in Flat Bandar Tasik Selatan–FBTS and Surya Magna–SM in Kuala Lumpur, Malaysia by simulation and measurement. This study discussed the four urban configurations, various thermal comfort indices, environmental parameters, which has some guiding significance for the urban design. Some specific comments are as follows:

(1) The Section 2: Literature Review is suggested to be merged with the section 1: Introduction. In addition, the descriptions from Line:228-243 could also be removed to the section 1.

(2) The superscript and subscript of some parameters in all equations in this manuscript should be normatively labelled.

(3) The sub-figure and their titles in Table 2 (Case 1) should be clearly presented.

(4) The test locations for the air temperature, relative humidity and air velocity in Fig. 2-7 were not clear.

(5) Line 264-266: How to determine the grid cells? Please add some more comparative investigation between the results from different grid numbers.

(6) Table 3: What the meaning of heat transmission walls (heat transmission roofs), albedo walls (albedo roofs)? Are they the heat convection coefficients and reflectivity of walls and roofs?

(7) The descriptions in the Figs. 10-13 should be discussed in the main text of the manuscript.

(8) Why are some values of the mean solar radiation in Table 4 so large? For instance, some values are 1204.90, 2077.23, 3075.72 W/m2.

(9) Line373-416: Are the descriptions in these four paragraphs relevant to the Table 4?

(10) The conclusions are tedious and should be simplified.

(11) It is suggested that the authors should carefully check the full-text language of this manuscript.

Author Response

(The authors gave the same response as above.)

Reviewer 4 Report

The article entitled The impacts of urban configurations on outdoor thermal perceptions: Case studies of Flat Bandar Tasik Selatan and Surya Magna in Kuala Lumpur, presents as general objective:

(a) To evaluate the impact of urban configurations on outdoor thermal comfort.

(b) To examine thermal perceptions and discuss the impact of urban forms on the perceptions.

(c) To discuss the applications and recommendations based on the outcomes of the research.

This is an important contribution to the understanding of human thermal comfort in a hot and humid climate where air humidity plays an important role in this process. It presents an important bibliographic review, which demonstrates the careful and up-to-date reading of the authors of the article. Modeling via ENVIMET is used to produce scenarios under two conditions:two empirical sites were evaluated for analysis. Firstly, the research  considered the scenario of empirical urban configurations with the canyon direction of East-West as well as in the North-South direction. Both sites are in a high-rise residential  buildings zone in Kuala Lumpur in Malaysia. Results indicate that it is a region of high daytime and nighttime temperatures. It is noteworthy that the response of the daily curve of air temperature does not present a symmetric curve to solar radiation. In general, the acronyms presented in the figures need to be defined, as for example, in figure 10 the term U. Results indicate that the orientation of buildings is an important element of the variation of climatic attributes and should be considered in the architectural design of these buildings. General comments: - article too long for the journal's standards, - necessary to include a climatic characterization of the study area (normal values ​​of temperature and precipitation in particular). - include a location map of the study area - from global to local.

Author Response

(The authors gave the same response as above.)

Round 2

Reviewer 1 Report

The authors have revised the paper extensively. While I have still doubts about your introduction and method.

1)You metioned about UHI in introduction. What is the relation of UHI and thermal comfort? As we know, UHI is an index to evaluate the temperature difference between city center and suburb, whihe UTCI and WGBT index is to evaluate thermal  comfort and thermal stress. Is it  neccessory  to combine these concepts together?

2)UTCI and WGBT are two different indices. Why you use the two different index together?

3)What is the local UTCI scale? It could affect the thermal acceptance profoundly. I suggest authors add some references, such as https://doi.org/10.3390/buildings12060720 ,https://doi.org/10.1016/j.buildenv.2020.106739

Author Response

3rd October 2022

Guest Editors

Buildings Journal

Dear Editors,

The authors would like to thank the reviewers for the careful review and their feedbacks which have helped to strengthen the quality of the manuscript.

The manuscript entitled ‘The impacts of urban configurations on outdoor thermal perceptions: Case studies of Flat Bandar Tasik Selatan and Surya Magna in Kuala Lumpur’ has been revised accordingly and changes are highlighted in red. Please find attached our response report to the Reviewers.

Thank you, and we look forward to receiving your reply.

Sincerely

Dr. Lin Yola

Senior Lecturer,

School of Strategic and Global Studies, Universitas Indonesia, Jakarta, Indonesia

[email protected]

Reviewer 3 Report

The revised manuscript could be accepted for the publication.

Author Response

(The authors gave the same response as above.)
